# A Deep Learning Approach to Organic Pollutants Classification Using Voltammetry

**DOI:** 10.3390/s22208032

**Published:** 2022-10-21

**Authors:** Mario Molinara, Rocco Cancelliere, Alessio Di Tinno, Luigi Ferrigno, Mikhail Shuba, Polina Kuzhir, Antonio Maffucci, Laura Micheli

**Affiliations:** 1Department of Electrical and Information Engineering, University of Cassino and Southern Lazio, 03043 Cassino, Italy; 2Department of Chemical Science and Technologies, University of Rome “Tor Vergata”, 00133 Rome, Italy; 3Center of Physical Science and Technologies, 10257 Vilnius, Lithuania; 4Institute of Photonics, Department of Physics and Mathematics, University of Eastern Finland, 80101 Joensuu, Finland; 5INFN, Italian National Institute for Nuclear Physics, 00044 Frascati, Italy

**Keywords:** carbon nanotubes, convolutional neural networks, pollutant detection, screen-printed electrodes, cyclic voltammetry

## Abstract

This paper proposes a deep leaning technique for accurate detection and reliable classification of organic pollutants in water. The pollutants are detected by means of cyclic voltammetry characterizations made by using low-cost disposable screen-printed electrodes. The paper demonstrates the possibility of strongly improving the detection of such platforms by modifying them with nanomaterials. The classification is addressed by using a deep learning approach with convolutional neural networks. To this end, the results of the voltammetry analysis are transformed into equivalent RGB images by means of Gramian angular field transformations. The proposed technique is applied to the detection and classification of hydroquinone and benzoquinone, which are particularly challenging since these two pollutants have a similar electroactivity and thus the voltammetry curves exhibit overlapping peaks. The modification of electrodes by carbon nanotubes improves the sensitivity of a factor of about ×25, whereas the convolution neural network after Gramian transformation correctly classifies 100% of the experiments.

## 1. Introduction

Fast and reliable detection and classification of organic pollutants in water is a major challenge in today’s society, addressing the goals of sustainable development [1]. Classical and well-assessed techniques such as those based on chromatographic and spectrophotometric analysis are known to provide reliable detection and classification (e.g., [2,3,4]). However, these methods require in-presence sample processing and high employment of reagents and time. For these reasons, recently, attention has been paid to electrochemical methods such as voltammetry (VA), with disposable screen-printed electrodes (SPEs) as sensing platforms. Indeed, these approaches have been shown to provide good sensing performance [5,6,7], while enabling the possibility of moving the analysis from the lab to in situ [8]. Furthermore, recently improved performance by modifying SPEs with 2D nanocarbon materials (e.g., carbon nanotubes or graphene nanoplatelets films [9,10,11]) was demonstrated.

Cyclic voltammetry (CV) is undoubtedly the most widely used technique for acquiring qualitative information about electrochemical behavior and characterization of target analyte, and it is often the first experiment performed in an electroanalytical study. CV is a powerful tool for the rapid determination of formal potentials, and detection of chemical reactions, preceding by following the electron transfer process and by evaluating the electron transfer kinetics [12,13]. Though CV is highly informative, its high background current imposes some limitations on its use in the quantitative determination of various molecules. To overcome the problem of low limit of detection, CV is usually complemented by more sensitive techniques, such as differential pulse voltammetry (DPV), square wave voltammetry (SWV), and amperometry [14,15].

However, when classification must be addressed, the task becomes challenging. Indeed, the identification of the pollutant is based on the position of the peaks in the CV voltammograms by a comparison with those of standards. Unfortunately, the SPE platforms can be highly sensitive to fabrication processes (printing procedures, inks, substrates), which can significantly affect their reproducibility; moreover, the uncertainty of the electrochemical measurements can be related to temperature variation, stability of the instrumentation, and errors of operators, influencing the signal/noise (S/N) ratio, and the reliability of the test results [16,17]. As a consequence, the shape of CV voltammograms is affected by issues such as broadening, asymmetry, and tailing peak. This condition is likely to lead to an important overlap of the peak currents, which significantly hinders the measurement of quantitative parameters correlated to it and the peak potential [18]. If this condition occurs in the case of the detection of different analytes with a similar footprint, their correct classification is hard to be achieved using of the standard approach of voltammograms post-processing. Therefore, intensive studies have been carried out in the last decades to provide solutions to this problem. For instance, in [19], a method for the separation and the improvement of the resolution of overlapping peaks (by decreasing the “background” noise and increasing the S/N) in cyclic voltammetry based on a semi-differential transformation algorithm is proposed.

Both classic machine learning [20] and deep learning [21] approaches for CV analysis have been applied in the last few years.

A deep learning approach was followed in [22,23], where the input was represented directly by the CV, as usual in this kind of approach. The limits of the latter are related to the impossibility of using 2D CNN, which has demonstrated in the last year very high performances [21]. In [24], a complete review of machine learning techniques applied to voltammetric electrochemical sensing is presented.

This paper proposes an innovative technique to assess a reliable classification from VA measurements, which is based on deep learning paradigms and on the curve to image transformation by using 2D CNN. The proposed approach is adopted to classify two organic substances in water, i.e., benzene metabolites benzoquinone (BQ) and hydroquinone (HQ). HQ and BQ are undoubtedly two of the most important water pollutants occurring in the environment because of anthropogenic processes (industrial production), as well as natural products from plants and animals [25,26]. Exposure to such substances is known to lead to serious health problems, so they are classified as harmful chemicals [27]. Moreover, their detection and classification through sensing systems based on SPEs is still challenging.

In addition, the detection of quinones via VA techniques is known to be demanding, due to the problem of the passivation of the electrodes, because they can be electropolymerized during the measurements, and a polymeric layer that prevents a regular redox process can coat the working electrode. Consequently, the SPE platform can be used only once (disposable) [28]. For this reason, alternative detection techniques such as electrical spectroscopy are investigated [29,30,31,32,33,34], based on the high sensitivity in the frequency of electrical permittivity of carbon nanomaterials [35]. This problem introduces an additional source of uncertainties in the measured data, due to the need to replace the SPE platform after each cycle, with a degradation of the measurement repeatability.

The deep learning approach proposed here is intended to improve electrochemical techniques such as VA and provide accurate classification of the pollutants starting from noisy data, coming from measurements done with different platforms at various times and by different operators.

To this end, the CV data are first transformed into “equivalent images” by using a Gramian angular field (GAF) technique [36]. Then, by applying a deep learning approach [37], a convolutional neural network (CNN) is trained to classify HQ and BQ, using ferricyanide of potassium (K₃[Fe(CN)₆]) as the electroactive reference molecule, obtaining a three-class classification problem

The paper is organized as follows. In Section 2, the CV measurement technique is reviewed, and details are provided on the data obtained by using several types of SPE platforms. Section 3 deals with the proposed deep learning approach. In Section 4, case studies are analyzed, and the classification result is discussed.

## 2. Experimental Characterization

### 2.1. Detection via Cyclic Voltammetry

The characterization technique adopted here is based on the use of low-cost disposable screen-printed electrodes (SPEs), which include working electrode (WE), counter electrode (CE), and reference electrode (RE), as shown in Figure 1a. In this paper, different electrochemical platforms are compared (all printed on polyethylene support), where the WE is either made by graphite ink (hereafter denoted as “bare electrode”) or modified by single or multiwalled carbon nanotube (S or MWCNT) films (denoted as “modified electrode”) to understand their electrochemical performances.

An example of a CV electrochemical characterization (Figure 1) using hydroquinone, as an elective compound, is carried out using several dilutions (Figure 1b) of this compound to build the calibration curve (Figure 1c) to understand the sensitivity of the SPE. The analyte solution (80 μL) is dropped onto the WE, whose electrical potential V (concerning the reference electrode) is linearly cycled *n* times between a maximum and a minimum value. The resulting electrical current I, flowing through the working and counter electrodes, is measured, leading to the final I-V curves with a characteristic duck-shaped plot profile (see Figure 1b).

Studying the current (I) vs. potential (V) voltammogram allows the detection of a specific analyte in solution since its electrochemical properties determine the specific redox potentials profile of the compound, used to identify it in a mixture of real samples. In Figure 1b, an example is given referring to the detection of HQ in fortified water (adding a known amount of HQ in a fixed volume of water) with a characteristic potential profile (two reduction peaks and one in oxidation).

In addition to the classification of the analyte, the I-V curves obtained at different concentration levels also allow “calibrating” the sensor, as shown in Figure 1c. Specifically, the peaks of the current for each cycle corresponding to the oxidation can be related to the pollutant concentration value, giving the response in Figure 1c. For instance, in this case, the calibration curve is quite linear in the range of interest (fitting parameters provided in inset).

The Randles–Sevcik equation [14] can be used to extrapolate important analytical parameters from the peak current value, e.g., the electrode surface (*A*) and the diffusion coefficient (*D*_0_):(1)$$I_{p}=\left(0.4463\right)nFAC\sqrt{\frac{nFvD_{0}}{RT}}$$
where v is the scan rate (Volt/s), *n* is the number of electrons involved in the process, *F* is the Faraday constant (1/mol), *T* is the temperature (K), and *R* is the universal gas constant (J/K mol).

The limits of CV characterization, discussed in the introduction, can lead to poor sensitivity (especially at low concentration rates) and/or unacceptable selectivity. An example is provided in Figure 2, which reports the measured peak potentials and the corresponding current values related to the peaks of the cyclic voltammograms obtained with HQ at the concentration of 5 mM with different platforms: bare SPEs (green), MWCNT SPEs (blue), and SWCNT SPEs (red). It can be highlighted that changing the platforms (based on SPE) leads to a huge variation in the measured potential and current peaks. In addition, even with the same platform (i.e., same colors in Figure 2), a sensible spread of the value (low reproducibility in terms of RSD) is observed from one measurement to another. Similar behavior is also obtained for the other pollutant (BQ).

This high variability of the data, together with similar behavior of the CV responses to the two pollutants, makes it extremely difficult for their resolution with these kinds of techniques.

A way to face the first issue is based on the modification of SPE using carbon nanomaterials, as briefly reviewed in the following subsection. The second issue will be addressed using the machine learning approach discussed in Section 3.

### 2.2. Carbon Nanotube Modified Platform

To improve the CV response of the SPE, the WE was modified by CNT films by drop-casting. Single-walled CNTs (Heji Inc., Hong Kong) and multiwalled CNTs (Nanointegris Technology Inc., Boisbriand, QC, Canada) were used. CNT thin films were produced using the filtration method. Briefly, 0.2 mg of CNT powder was dispersed for 1 h in a 1% aqueous solution of sodium dodecyl sulfate in a 200 W ultrasonic bath at 44 kHz. To remove CNT agglomerates, the suspension was centrifuged at 8000 *g* for 20 min, and then filtered through a cellulose-ester membrane (Millipore, 0.22 µm pore size). As a result, SWCNTs and MWCNTs form thin—0.2 and 1 μm—films on the top of filter paper. To remove surfactant, CNT films on the filter paper were washed at 80 °C and then dried in air overnight.

To transfer on the SPEs, CNT films on the filter paper were put into water. The water was substituted with acetone to dissolve the filter paper, and then the acetone was replaced with ethanol. After its removal, the wet films were transferred to the WE and dried finally ready to be used.

The images obtained with scanning electron microscopy (SEM, see Figure 3) demonstrate a larger density of SWCNT than MWCNT film.

It is assumed that the large density of SWNT film prevented a complete removal of the surfactant from its surface. Raman spectra of CNTs (see Figure 4) show a low (0.08) and a high (0.86) ratio of D-mode to G-mode intensities for SWCNT and MWCNT films, respectively.

This indicates a low density of defect in the crystalline structure of SWCNTs compared to that of MWCNTs.

### 2.3. Characterization of Hydroquinone and Benzoquinone and Dataset Generation

The characterization of HQ and BQ was carried out by using both bare and modified SPEs. Potassium ferri/ferrocyanide [Fe(CN)_6_]^4−/3−^ (hereafter PF) was also analyzed as a reference electroactive species. All reagents from commercial sources were of analytical grade. Potassium ferri/ferrocyanide, p-benzoquinone, and hydroquinone were purchased from Sigma-Aldrich (Steinheim, Germany). The buffer solution used is a 0.05 M phosphate buffer saline (PBS), 0.1 M KCl, pH = 7.4.

CV was performed using a Palmesens4™ portable potentiostat system (Palmsens, Houten, The Netherlands) as an analytical tool and in-house produced screen-printed electrodes (SPEs) as a transducer.

An experiment of a simultaneous determination of the above three species (BQ, HQ, and PF) by using CV was carried out, referring to the concentration of 10 mM, and the use of bare electrodes. The results are shown in Figure 5, which reports the combined cycles (Figure 5a) and the separate ones (Figure 5b–d). These results evidence the challenge of correctly classifying the compounds.

The generated dataset refers to the CV characterization of HQ, BQ, and PF in water solution, with three different platforms: bare, SWCNT, and MWCNT modified SPEs. Three cycles were measured for each combination of compounds/platform/concentration, thus obtaining a dataset with a total of 291 experimental CV curves.

Examples of CV cycles are provided in Figure 6 for HQ, characterized by bare and CNT-modified SPEs. Table 1 quantifies the main performance parameters, such as the limit of detection (LOD) and reproducibility (RDS%). These results highlight improved performance obtained, once the SPEs are modified by carbon nanotubes, as modification results in a much lower LOD (more than one order of magnitude for MWCNT) and better reproducibility (lowering RDS% by a factor of two).

## 3. Classification via Machine Learning

### 3.1. Gramian Angular Fields Transformations

The first step of the proposed approach consists of mapping any I-V cycle obtained by CV into an equivalent red-green-blue (RGB) image. To this end, the Gramian angular fields (GAF) transformation was used [35], which is here briefly recalled. Starting from a time series $X=\left\{x_{1},x_{2},\;\ldots ,\;x_{n}\right\}$ of *n* real-valued observations, the vector X is normalized in such a way that all its values fall into the interval [−1,1]:(2)$${\tilde{x}}_{i}=\frac{\left(x_{i}-\max(X)+\left(x_{i}-\min(X)\right)\right)}{\max\left(X\right)-\min\left(X\right)}$$

The re-scaled vector $\tilde{X}$ is represented in polar coordinates by associating the value to the angular cosine, ${Ø}_{i}$, and the time instant to the radius, r
(3)$$\{\begin{array}{c} {Ø}_{i}=arcos\left({\tilde{x}}_{i}\right),\;-1\leq {\tilde{x}}_{i}\leq +1,\;{\tilde{x}}_{i}\in \tilde{X}\\ r=\frac{t_{i}}{N},\;i\in N \end{array}$$

Here, $t_{i}$ is the time stamp and N is a constant factor to regularize the span of the polar coordinate system.

This transformation has two important properties: (i) it is bijective as cos(Ø) is monotonic when Ø∈[0,π]; (ii) as opposed to Cartesian coordinates, polar coordinates preserve absolute temporal relations.

The Gramian summation angular field (GASF) and Gramian difference angular field (GADF) are defined as follows:(4)$$\;\;\;\;\;\;\;\;\;\;\;\;\;\{\begin{array}{c} GASF=\left[\cos\left({Ø}_{i}+{Ø}_{j}\right)\right]\\ GADF=\left[\sin\left({Ø}_{i}-{Ø}_{j}\right)\right] \end{array}$$

By using the above transformations, each I-V cycle was mapped into an image, as shown in Figure 7. Specifically, the GASF related to the potential was transformed in the red (R) color plane, and the GADF and GASF related to the current into the green (G) and blue (B) planes, respectively. Examples of generated RGB images are provided in Figure 8, referring to the detection of BQ in water solution.

### 3.2. Deep Learning Model

Once all the 291 cycles have been mapped into 291 RGB images, a suitable deep neural network (DNN) can be trained to classify them [27].

In this paper, we adopted a convolutional neural network (CNN), a category of DNN inspired by biological processes [38,39] as the model of connectivity between neurons recalls the organization of the animal visual cortex. Individual cortical neurons respond to stimuli only in a narrow visual field region known as the receptive field. The receptive fields of different neurons partially overlap to cover the entire visible area.

CNN uses relatively little preprocessing compared to other image classification algorithms. This means that the network learns to optimize filters (or kernels) through machine learning, whereas in traditional algorithms, these filters are designed manually. This independence from prior knowledge and human intervention in feature extraction is a great advantage when using such an approach.

Table 2 reports the characteristics of the relatively simple CNN designed for our purpose. The CNN contains five blocks with a 2D convolution (kernel size 3 × 3), a max-pooling (stride of 2), and two fully connected layers, with a hidden layer of size 64, a dropout of 0.5, and an output layer of size 3. Specifically, Table 2 reports the number of parameters layer by layer, with a total amount of 51,799. The dropout is typically used to reduce overfitting on the training set.

This model was finally chosen after a preliminary experimental phase assuming as a goal the minimization of the complexity of the network.

The selected network has a straightforward structure with a sequence of convolutional and max-pooling layers, a flattened layer, and two dense and fully connected layers for the classification, with a dropout to limit overfitting on the training set.

### 3.3. Dataset

As described in Section 3, the dataset generated by the CV experiments contains a total of 291 experimental points, referring to the characterization of PF, HQ, and BQ in water by means of three different platforms: bare SPEs, SWCNT, and MWCNT modified SPEs. Table 3 provides the complete description of the dataset.

The 291 available experiments were randomly assigned to the training, validation, and test sets in a ratio of 75% (227 images), 10% (29 images) and 15% (35 images), respectively.

To augment the statistical significance of the experiments, this subdivision was randomly repeated five times (5-fold cross-validation), obtaining a different train/validation/test set each time. For each of these subdivisions, a model was trained and evaluated.

## 4. Results and Discussion

The result of applying the GAF transformation to the dataset is an impressive stabilization of the pattern associated with any single test (detection of a given substance at a given concentration). This is a consequence of a dramatic reduction of the uncertainty related to using different platforms at different times for the given test. To show it, let us refer to the test case analyzed in Figure 2, i.e., the CV response associated with the detection of HQ at a concentration of 5 mM.

Once all the I-V curves related to this case are transformed into equivalent RGB images by using GAF, we can compute for each pixel and each color the average and standard deviation values associated with experiments available for the above test case. These values are shown in Table 4 for each color: the ratio STD/average ranges from 0.23 to 10.5%.

The subsequent deep learning processing was performed by using an Intel Core i7-7700 CPU@3.60GHz, 256GB of RAM with a GPU Titan Xp. As a deep learning framework, we used Keras version 2.4.0 with TensorFlow version 2.4.0 as the back end [40,41].

During the training of the network, the following hyperparameters were used:Epochs: 400Patience: 100Optimizer: Stochastic Gradient DescentLearning rate: 0.0001Momentum: 0.9Loss: categorical cross-entropyMetrics: accuracyBatch size: 16

At the end of each epoch, the loss was evaluated on the validation set to save the model with the best performance, avoiding overfitting. An early stopping policy was implemented with a patience of 100 to stop the learning phase if the loss does not improve for some time. Figure 9 and Figure 10 report examples of the accuracy and loss curve during the training phase on the first fold, evaluated on the training and the validation set at the end of each epoch.

In these curves, it is possible to observe that the loss on the validation set is lower than the loss on the training set; at the same time, it is possible to observe that the accuracy on the validation set is greater than the accuracy on the training set. This can happen in the presence of a dropout layer, where during training, a percentage of the features is set to zero (with a 50% of probability because we adopted a dropout of 0.5). During the validation, all features are used; consequently, the model is more robust, leading to higher accuracy.

In the loss curve (Figure 10), a plateau is reached at epoch 1100; after this epoch the loss evaluated on the validation set does not decrease for at least 100 epochs, so determining the stop of the training phase. The model saved at epoch 1100 became the best model found and was then used during the test phase.

The entire training phase on a Dell laptop with Core i7 as a processor, 32GB of RAM, and an NVIDIA 3060 GPU with 6 GB of dedicated memory employed around 1 h to reach the convergence ad epoch 1200.

Global performances were evaluated in terms of accuracy (5) and confusion matrix (CM).
(5)$$Accuracy=\frac{Correctly\;Classified\;Samples}{Total\;Samples}$$

A CM summarizes the results of the testing phase on different classes. The *CM_ij_* element represents the percentage of elements labeled as class *i* and predicted as class *j*. The ideal case is represented by a diagonal matrix, which means that all samples for each class were correctly classified. Furthermore, the global accuracy can be evaluated as the ratio between the CM trace (sum of the correctly classified) and the sum of all CM values.

The results of the 5-fold repetitions of the CNN training are reported in Figure 11 in terms of mean values.

It can be noticed that the confusion matrix is fully diagonal, hence indicating that the proposed network generates an accuracy of 100% in all five repetitions of the experiments. Consequently, Figure 11 does not report the standard deviation along the experiments, given that its values are equal to zero.

Figure 12 reports the global overview of the system after its training: the input is the entire curve, this curve is transformed via GAF in an RGB image, and the trained CNN classifies the image into one of three classes (BQ, HQ, or FP).

The entire process on a standard machine such as a laptop with Core i7 of 11th generation takes less than 1 s for the entire classification process, as reported in Figure 12. This time is adequate for an online detection of these substances.

## 5. Conclusions

The paper proposes a further step toward the realization of a low-cost AI-based embedded sensor for the detection and classification of organic pollutants in water. The proposed solution is based on suitable screen-printed electrodes connected to measurement micro-platforms capable of performing cyclic voltammetry tests and embedded with an innovative deep-learning algorithm for classification and detection. In detail, the paper is mainly focused on the optimization of the classification task that is executed with convolutional neural networks. The main novelty of the paper is the innovative use of Gramian angular fields transformations to transform in suitable RGB image data coming from voltammetry tests. To demonstrate the goodness of the solution, two challenging pollutants, i.e., hydroquinone and benzoquinone, that have very similar electroactivity and consequently very similar voltametric footprint, were considered. In addition, results coming from different types of screen-printed electrodes were considered. In this way, the obtained results are not related to a specific sensor but are a feature of the proposed platform.

Obtained results show that this preliminary conditioning of the measurement information allows us to deeply improve the performance of the convolutional neural networks allowing us to reach a classification accuracy of 100%.

Future work will be carried out in two directions: (a) to embed the classification capability in the measurement micro-platform and (b) to increase the number of pollutants and screen-printed electrodes to give more generalization capabilities to the developed platform.

## Figures and Tables

**Figure 1 sensors-22-08032-f001:**
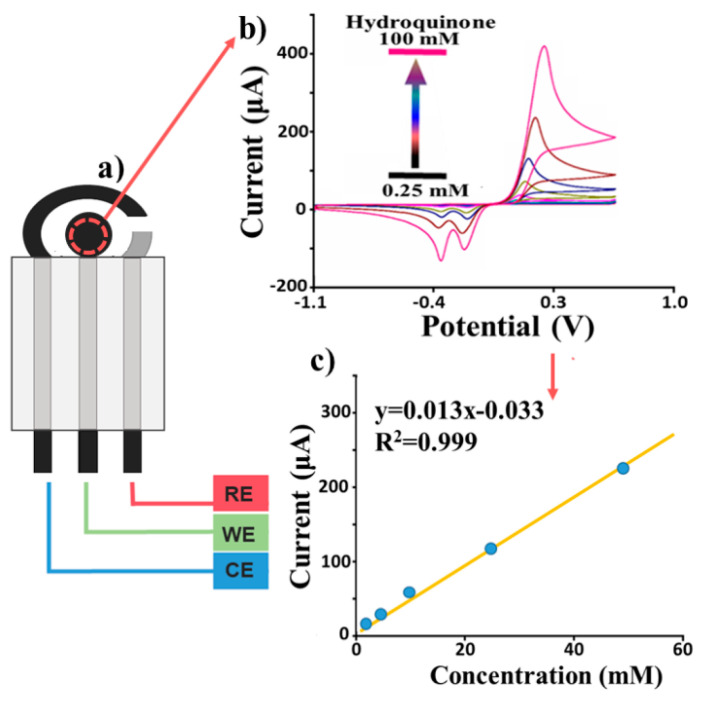
(**a**) A conceptual scheme of a screen-printed electrode; (**b**) example of current-potential curves as the output of a cyclic voltammetry for different values of the analyte concentration (here hydroquinone); (**c**) extrapolated calibration curve (blue dots: experimental current peaks).

**Figure 2 sensors-22-08032-f002:**
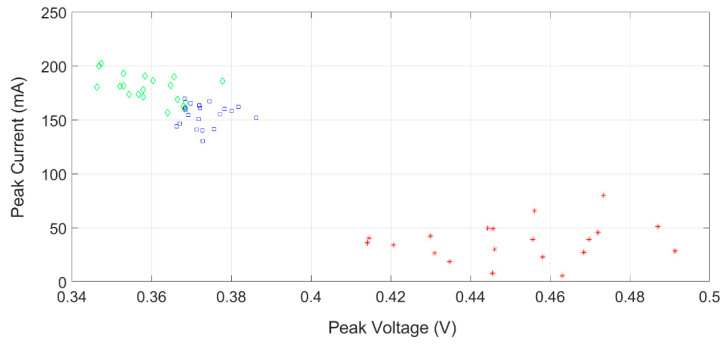
Peak potential and currents measured on several cyclic voltammograms executed on the hydroquinone pollutant (5 mM), with different platforms: Bare SPEs (green), MWCNT SPEs (blue), and SWCNT SPEs (red).

**Figure 3 sensors-22-08032-f003:**
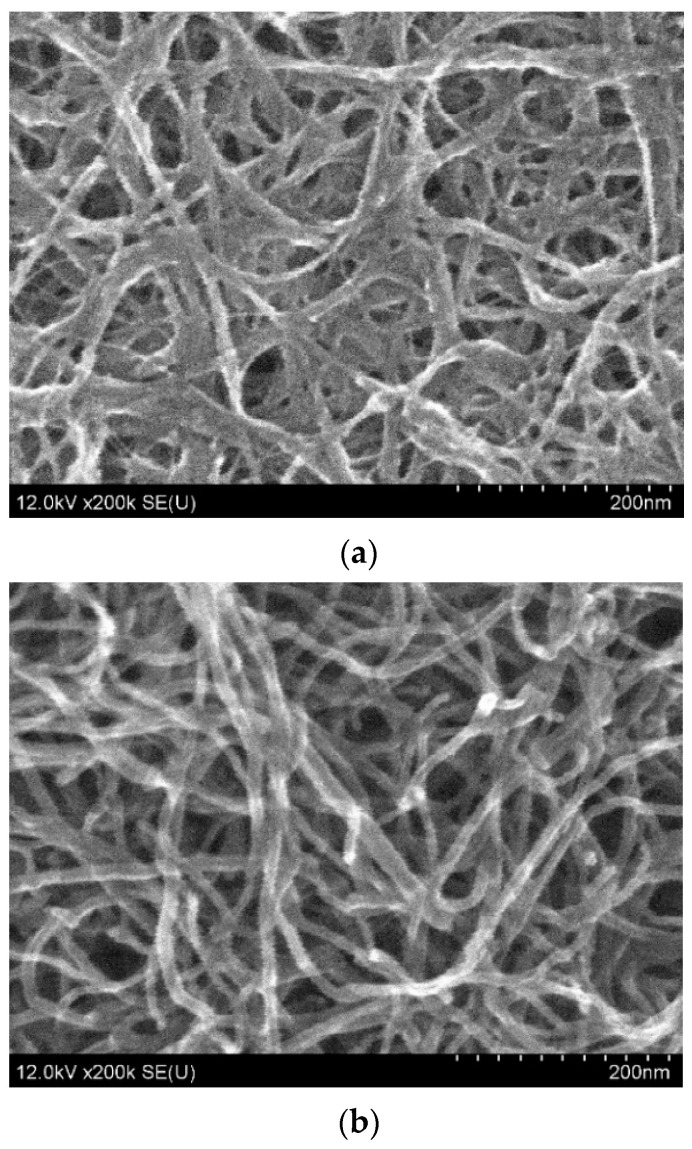
Scanning electron microscope (SEM) images of the produced (**a**) SWCNT and (**b**) MWCNT.

**Figure 4 sensors-22-08032-f004:**
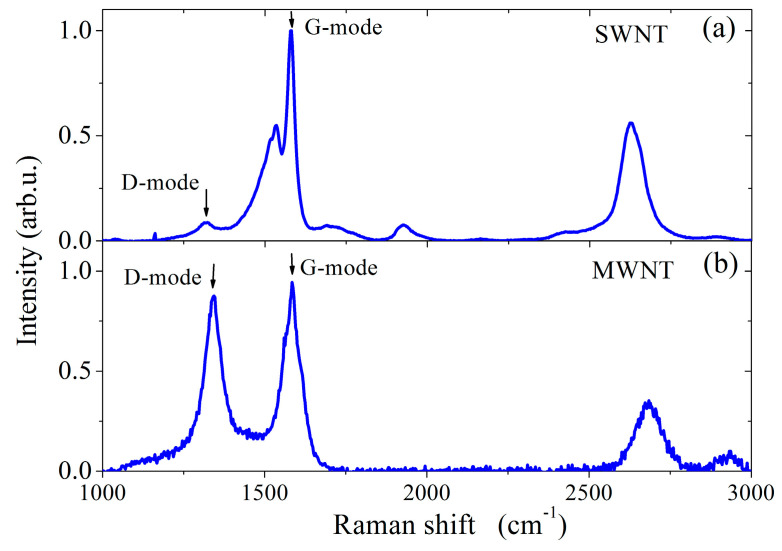
Raman spectra of (**a**) single-walled and (**b**) multiwalled CNTs.

**Figure 5 sensors-22-08032-f005:**
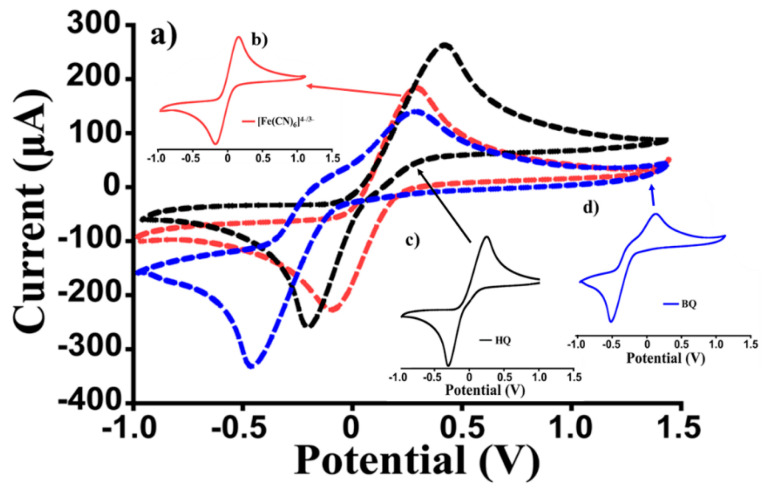
Comparison of the cyclic voltammograms; (**a**) simultaneous determination of the 3 analytes; (**b**–**d**) PF, HQ, and BQ were analyzed separately.

**Figure 6 sensors-22-08032-f006:**
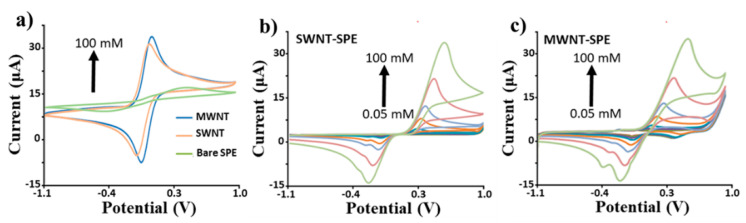
CV voltammograms obtained with different platforms (SWCNTs, MWCNTs, and bare SPEs), referred to: (**a**) the same concentration of HQ (5 mM), and (**b**,**c**) several HQ concentrations (from 0.1 µM to 1 mM).

**Figure 7 sensors-22-08032-f007:**
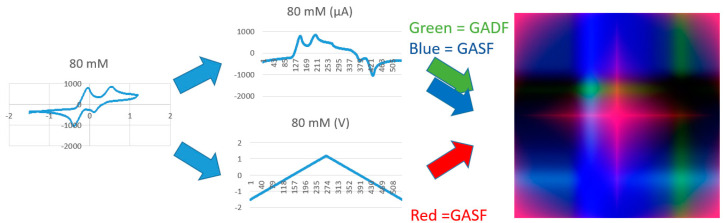
Gramian angular field (GAF) transformation of an I–V cycle into an RGB image.

**Figure 8 sensors-22-08032-f008:**
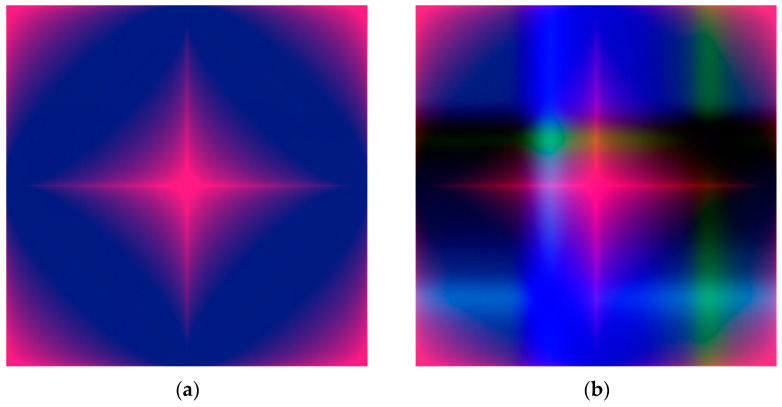
RGB images generated by GAF transformation of CV cycles, for the detection of: (**a**) Potassium ferricyanide (PF) in water, 0.25 mM, bare SPE; (**b**) potassium ferricyanide (PF) in water, 100 mM, bare SPE; (**c**) benzoquinone (BQ) in water, 2.5 mM, SWCNT SPE; (**d**) benzoquinone (BQ) in water, 80 mM, SWCNT SPE.

**Figure 9 sensors-22-08032-f009:**
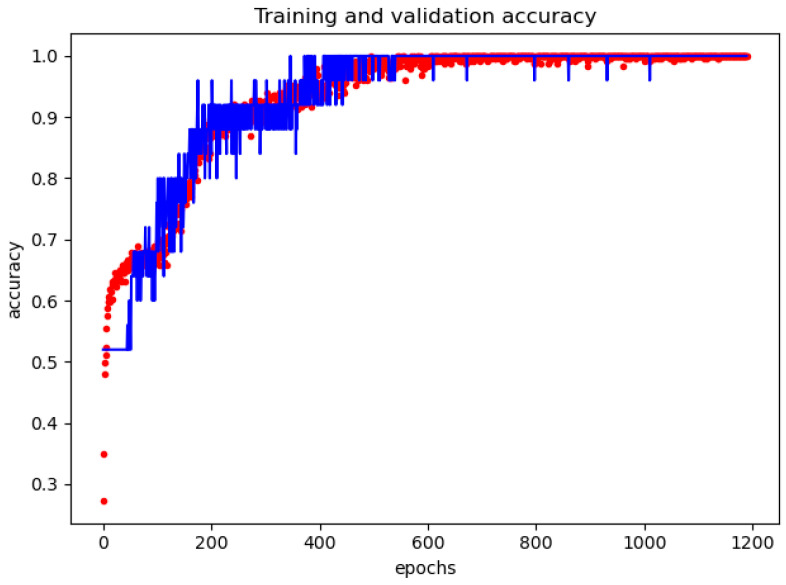
Evolution with the epochs of the accuracy during the training phase on the training set (red dotted curve) and the validation set (blue solid line) for the first fold.

**Figure 10 sensors-22-08032-f010:**
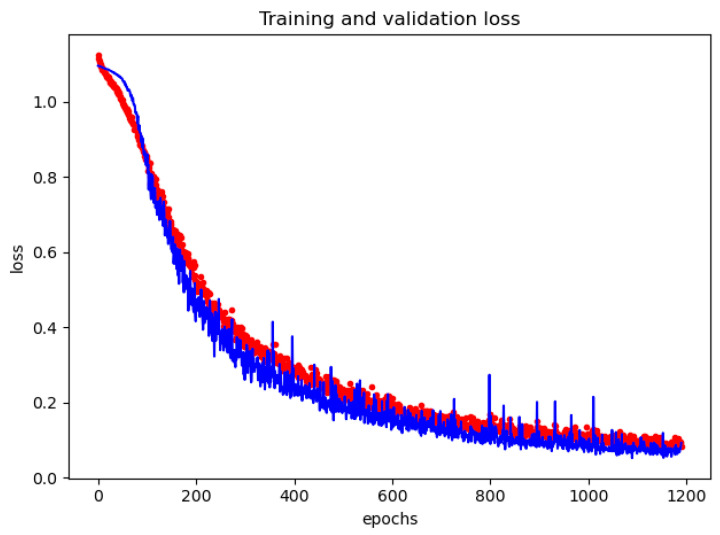
Evolution with the epochs of the loss during the training phase on the training set (red dotted curve) and validation set (blue solid curve) for the first fold.

**Figure 11 sensors-22-08032-f011:**
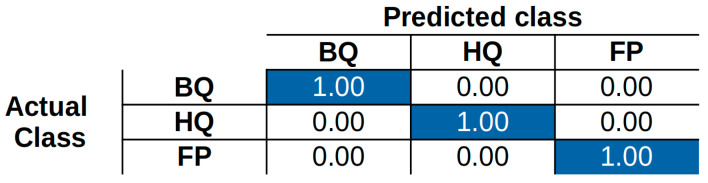
Confusion matrix evaluated on the test set.

**Figure 12 sensors-22-08032-f012:**
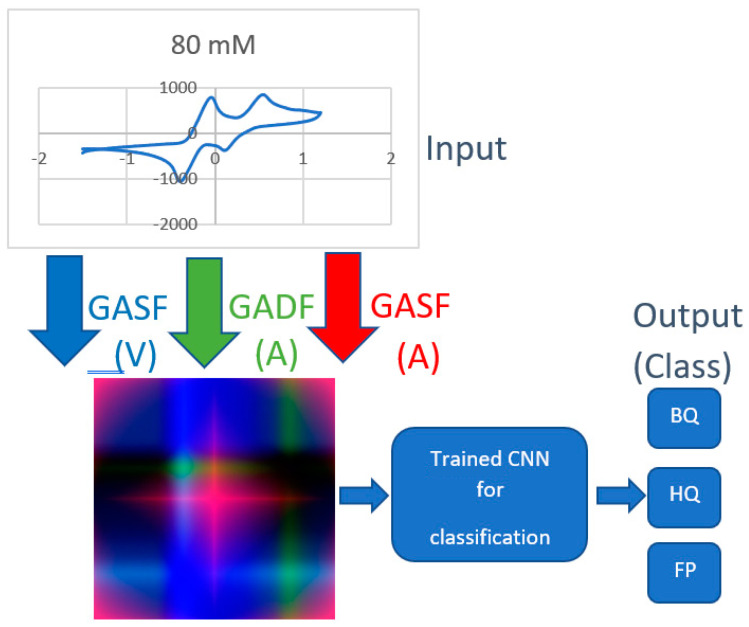
A global overview of the entire workflow of the proposed system.

**Table 1 sensors-22-08032-t001:** Limit of detection (lod) and reproducibility (rds%) for cv determination of hq in water, with different platforms.

Parameter/Electrode	Bare	SWCNT	MWCNT
LOD (μM)	334.5	80.3	13.7
Reproducibility (RSD%)	17	8	9

**Table 2 sensors-22-08032-t002:** Characteristics of the designed cnn.

Layer (Type)	Output Shape	Size	Param #
input	(None, 224, 224, 3)		
conv2d (Conv2D)	(None, 222, 222, 48)	3	1.344
max_pooling2d (MaxPooling2D)	(None, 111, 111, 48)	2	0
conv2d (Conv2D)	(None, 109, 109, 48)	3	20.784
max_pooling2d (MaxPooling2D)	(None, 54, 54, 48)	2	0
conv2d (Conv2D)	(None, 52, 52, 32)	3	13.856
max_pooling2d (MaxPooling2D)	(None, 26, 26, 32)	2	0
conv2d (Conv2D)	(None, 24, 24, 32)	3	9.248
max_pooling2d (MaxPooling2D)	(None, 12, 12, 32)	2	0
conv2d (Conv2D)	(None, 10, 10, 16)	3	4.624
max_pooling2d (MaxPooling2D)	(None, 5, 5, 16)	2	0
conv2d (Conv2D)	(None, 3, 3, 8)	3	1.160
max_pooling2d (MaxPooling2D)	(None, 1, 1, 8)	2	0
Flatten	(None, 8)		0
Dense	(None, 64)		576
Dropout (0.5)	(None, 64)		0
Dense	(None, 3)		195
Batch Normalization	(None, 3)		12
Activation Softmax	(None, 3)		0
	Total param #		51,799

**Table 3 sensors-22-08032-t003:** Available dataset after cv characterization.

Benzoquinone		Sensors			
		Bare	MWCNT	SWCNT	
	Number of voltammetry cycles	2	3	3	
	Concentrations (mM)		80	80	
		50	50	50	
		25	25	25	
		12.5	12.5	12.5	
		5	5	5.1	
		2.5	2.5	5	
		1			
					Total images:
	Number of images	12	18	18	48
Hydroquinone		Sensors			
		Bare	MWCNT	SWCNT	
	Number of voltammetry cycles	3	3	3	
	Concentrations (mM)	100	100	100	
		50	50	50	
		25	25	25	
		12.5	12.5	12.5	
		5	5	5	
		2.5	2.5	2.5	
		1	1	1	
		0.5	0.5	0.5	
		0.25	0.25	0.25	
					Total images:
	Number of images	27	27	27	81
Potassium ferricyanide		Sensors			
		Bare	MWCNT	SWCNT	
	Number of voltammetry cycles	6	6	6	
	Concentrations (mM)	100	100	100	
		50	50	50	
		25	25	25	
		12.5	12.5	12.5	
		5	5	5	
		2.5	2.5	2.5	
		1	1	1	
		0.5	0.5	0.5	
		0.25	0.25	0.25	
					Total images:
	Number of images	54	54	54	162

**Table 4 sensors-22-08032-t004:** Average and standard deviation values for the rgb image obtained by gaf transformation.

	*R*	*G*	*B*
Average	2.4596	2.7999	2.4832
STD	0.1590	0.2940	0.0057

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
