# Peer review of "A Deep Learning Approach to Organic Pollutants Classification Using Voltammetry"

_sensors, 2022, doi:10.3390/s22208032_

Round 1

Reviewer 1 Report

The paper is well written and combines Low-cost screen-printed electrodes (modified and unmodified with nanomaterials) with the Deep Leaning approach to accurately detect and classify organic pollutants.

However, concerns shall be handled before it is considered for publication. If the following problems are well-addressed, this reviewer believes that this paper's essential contribution is important for classifying organic pollutants, especially Hydroquinone and Benzoquinone. These concerns are listed below.

Experimental data

1. The graphic abscissa and ordinate values in Figure 5 (a) do not correspond to those in (b-d).

2. In Figure 6 (b,c), the concentration of HQ is labeled 0.05 mM-100 mM, but at the bottom, it is labeled from 0.1 µM to 1 mM.

Language Description:

3. Throughout the article, 'Multi-Walled Carbon Nanotube' has two abbreviations. MWCNT is used at the beginning and in some tables, but in the following texts, MWNT replaces MWCNT, and so does SWNT.

4. Similar to the previous one, in Figure 6, there are various words that represent the same platform, such as MWNT, MWNT-SPE, and MWNTs.

5. The passage above Figure 7 indicates that GADF is related to green, but Figure 7 shows GREEN=GASF. There is a problem with illustrative inconsistencies.

Layout:

6. To see the picture more conveniently, figure 4 should be placed in Section 2.2 instead of 2.3.

Grammars

7.Line21:"an accurate detection" should be "accurate detection", because detection is an uncountable noun here.

8.Line22: "leaning" should be changed to "learning."

9.Line166: "20 min" should be changed to "20 mins".

10.Line275, 278: According to the usage of other tables, 'Table II' should be changed to 'Table 2'.

References

11. Of the 38 references, 19 were published more than five years ago.

Author Response

The paper is well written and combines Low-cost screen-printed electrodes (modified and unmodified with nanomaterials) with the Deep Leaning approach to accurately detect and classify organic pollutants.

However, concerns shall be handled before it is considered for publication. If the following problems are well-addressed, this reviewer believes that this paper's essential contribution is important for classifying organic pollutants, especially Hydroquinone and Benzoquinone. These concerns are listed below.

 We would also like to thank the referee for his time reviewing our manuscript and their detailed reports and valuable suggestions.

Experimental data

  1. The graphic abscissa and ordinate values in Figure 5 (a) do not correspond to those in (b-d).

Response 1:

Thank you for this question. The graphic abscissa values in fig.5 b, c and d are different to fig. 5a in order to amplify the current answer of the single compounds. The graphic ordinate values are the same in all, probably the fact that fig. 5a is enlarged compared to the others, it makes it look different.

  1. In Figure 6 (b,c), the concentration of HQ is labeled 0.05 mM-100 mM, but at the bottom, it is labeled from 0.1 µM to 1 mM.

Response 2:

Thanks for underlining this mistake. The correct concentration is 0.1 µM -1mM.

Language Description:

  1. Throughout the article, 'Multi-Walled Carbon Nanotube' has two abbreviations. MWCNT is used at the beginning and in some tables, but in the following texts, MWNT replaces MWCNT, and so does SWNT.

Response 3: Thanks for underlining this mistake. Fixed

  1. Similar to the previous one, in Figure 6, there are various words that represent the same platform, such as MWNT, MWNT-SPE, and MWNTs.

Response 4: Thanks for underlining this mistake. Fixed

  1. The passage above Figure 7 indicates that GADF is related to green, but Figure 7 shows GREEN=GASF. There is a problem with illustrative inconsistencies.

Response 5: Thanks for underlining this mistake. Fixed

Layout:

  1. To see the picture more conveniently, figure 4 should be placed in Section 2.2 instead of 2.3.

 Response 6: Thanks for underlining this mistake. Fixed

Grammars

7.Line21:"an accurate detection" should be "accurate detection", because detection is an uncountable noun here.

Response 7: Thanks for underlining this mistake. Fixed

8.Line22: "leaning" should be changed to "learning."

Response 8: Thanks for underlining this mistake. Fixed

9.Line166: "20 min" should be changed to "20 mins".

Response 9: Thanks for underlining this mistake. Fixed

10.Line275, 278: According to the usage of other tables, 'Table II' should be changed to 'Table 2'.

 Response 10: Thanks for underlining this mistake. Fixed

References

  1. Of the 38 references, 19 were published more than five years ago.

Response 11:

Thank you for this question.

New references published in recent years have been added to the introduction and reported in red in the reference section.

Reviewer 2 Report

 1.     Abstract should be pruned with the solution process. However, the significant conclusions of the paper must be briefly mentioned at the last paragraph of the abstract. Need modification.

2.     Need brief Solution Methodology, working flow chart of the process.

3.     Fig. 1 and 2 descriptions are not clear, and need more details.

4.     Describe the convergence analysis.

5.     More explanation is needed for Fig. 5 (a). The comparison shows larger fluctuation. Need to recheck.

6.     The conclusion section needs modification.

7.     To fortify the literature, the recent and relevant articles should add and discuss within the text.

·        https://doi.org/10.1016/j.apt.2022.103551

Author Response

We would also like to thank the referee for his time reviewing our manuscript and their detailed reports and valuable suggestions.

  1. Abstract should be pruned with the solution process. However, the significant conclusions of the paper must be briefly mentioned at the last paragraph of the abstract. Need modification.

Response 1:

Thank you for this question, the abstract has been rewritten following the reviewer’s suggestion.

This paper proposes a Deep Leaning technique for an accurate detection and a reliable classification of organic pollutants in water. The pollutants are detected by means of Cyclic Voltammetry characterizations made by using low-cost disposable screen printed electrodes. The paper demonstrates the possibility of strongly improving the detection of such platforms, by modifying them with nanomaterials. The classification is instead addressed by using a Deep Learning approach with convolutional neural networks. To this end, the results of the voltammetry analysis are into equivalent images by means of Gramian Angular Transformations. The proposed technique is applied to the detection and classification of Hydroquinone and Benzoquinone, which are particularly challenging since these two pollutants have a similar electroactivity and thus the voltammetry curves exhibit overlapping peaks. The modification of electrodes made by using carbon nanotubes is shown to improve the sensitivity of a factor of about x25, whereas the convolution neural network after Gramian transformation is demonstrated to correctly classify 100% of the experiments.

  1. Need brief Solution Methodology, working flow chart of the process.

Response 2:

Thanks for underlining this mistake. Fixed

  1. Fig. 1 and 2 descriptions are not clear, and need more details.

Response 3:

Thank you for this question. The descriptions have been completed with more details.

  1. Describe the convergence analysis.

Response 4:

Thank you for this question. More details about convergences have been provided.

In the loss curve (Figure 10), a plateau is reached at epoch 1100; after this epoch the loss evaluated on the validation set doesn’t decrease for at least 100 epochs, so determining the stop of the training phase. The model saved at epoch 1100 became the best model found and was then used during the test phase.

  1. More explanation is needed for Fig. 5 (a). The comparison shows larger fluctuation. Need to recheck.

Response 5:

Thank you for the question. There is a huge difference between the measurements conducted on a single analyte and a mix of them. Indeed, by analysing them simultaneously, an incredible change in the background current is obtained; hence, also the current peaks are slightly changed. It is precisely why the use of machine learning is necessary for this condition. HQ, BQ, and the couple Fe2+/Fe3+ present peaks in very similar positions (Potential), therefore peaks overlap usually occur in this condition. This is ascribable to the sensitivity of the technique (μmolar range), indeed the CV with its triangular potential is not able to adequately separate the peaks related to different analytes. This paper aims to demonstrate that machine learning is able to solve this problem. Thus, allowing a semi-quantitative characterisation through CV.

  1. The conclusion section needs modification.

Response 6:

Thank you for this question. The conclusion has been completely rewritten.

The paper proposes a further step towards the realization of a low-cost AI-based embedded sensor for the detection and classification of organic pollutants in water. The proposed solution is based on suitable screen printed electrodes connected on measurement micro-platforms capable of performing cyclic voltammetry tests and embedded with an innovative deep learning algorithm for classification and detection. In detail, the paper is mainly focused on the optimization of the classification task that is executed with convolutional neural networks. The main novelty of the paper is the innovative use of Gramian Angular Fields transformations to transform in suitable RGB images data coming from voltammetry tests. To demonstrate the goodness of the solution, two challenging pollutants, i.e. Hydroquinone and Benzoquinone, that have very similar electroactivity and consequently very similar voltammetric footprint have been considered. In addition, results coming from different types of screen printed electrodes have been considered. In this way the obtained results are not related to a specific sensor but are a feature of the proposed platform.

Obtained results show as this preliminar conditioning of the measurement information allows to deeply improve the performance of the convolutional neural networks allowing to reach a classification accuracy of 100%. 

Future work will be carried out along two directions: a) to embed the classification capability in the measurement micro-platform and b) to increase the number of pollutants and screen printed electrodes to give more generalization capabilities to the developed platform.

  1. To fortify the literature, the recent and relevant articles should add and discuss within the text.

Response 7:

Thank you for this question.

New references published in recent years have been added to the introduction and reported in red in the reference section.

Reviewer 3 Report

The authors present a study on sensor technology for organic pollutants based on cyclic voltammetry. The application with a successful deep learning approach is well described in the paper. I recommend publication after minor revision.

1. The characteristics of the convolutional neural network (line 267) could be described in more detail. Although I have to admit I cannot judge if the design chosen here is "obvious" or if it needs more explanation.

2. How long does the deep learning processing take (line 313)? Does the computing time limit the application of the introduced technique?

3. I noticed following typos: line 22 "deep leaning", and line 143 "vs-1"

Author Response

The authors present a study on sensor technology for organic pollutants based on cyclic voltammetry. The application with a successful deep learning approach is well described in the paper. I recommend publication after minor revision.

We want to thank the referee for his time reviewing our manuscript and their detailed reports and valuable suggestions.

  1. The characteristics of the convolutional neural network (line 267) could be described in more detail. Although I have to admit I cannot judge if the design chosen here is "obvious" or if it needs more explanation.

Response 1:

Thank you for this question.

The network is a typical and simple CNN, but we agree with the reviewer that some comments are needed, so we added the following sentence:

The selected network has a very simple structure with a sequence of convolutional and max-pooling layers, a flattened layer, and two dense and fully connected layers for the classification, with a dropout to limit overfitting on the training set.            

  1. How long does the deep learning processing take (line 313)? Does the computing time limit the application of the introduced technique?

Response 2:

Thank you for this question.

Both the training time and the inference time (tes on a single curve) has been reported into the paper:

The entire training phase on a Dell laptop with Core i7 as a processor, 32GB of RAM, and a NVIDIA 3060 GPU with 6 GB of dedicated memory employ around 1 hour to reach the convergence ad epoch 1200.

The entire inference process, as reported in Figure 12, on a standard machine like a Laptop with Core i7 of 11th generation, employs less than 1 second to be completed. This time is adequate for an online detection of these substances.

  1. I noticed following typos: line 22 "deep leaning", and line 143 "vs-1"

Response 3: Thanks for underlining this mistake. Line 22 is fixed. Line 143 is fixed.